# Veterinary trypanocidal benzoxaboroles are peptidase-activated prodrugs

**Federica Giordani**[1⊙], **Daniel Paape**[1⊙], **Isabel M. Vincent**[1], **Andrew W. Pountain**[1], **Fernando Fernández-Cortés**[1], **Eva Rico**[2], **Ning Zhang**[2], **Liam J. Morrison**[3], **Yvonne Freund**[4], **Michael J. Witty**[5], **Rosemary Peter**[5], **Darren Y. Edwards**[2], **Jonathan M. Wilkes**[1], **Justin J. J. van der Hooft**[6,7], **Clément Regnault**[6], **Kevin D. Read**[2], **David Horn**[2], **Mark C. Field**[2,8], **Michael P. Barrett**[1,6]*

**1** Wellcome Centre for Integrative Parasitology, Institute of Infection, Immunity and Inflammation, University of Glasgow, Glasgow, United Kingdom, **2** Wellcome Centre for Anti-Infectives Research, School of Life Sciences, University of Dundee, Dundee, United Kingdom, **3** Roslin Institute, Royal (Dick) School of Veterinary Studies, University of Edinburgh, Midlothian, United Kingdom, **4** Anacor Pharmaceuticals, Inc., Palo Alto, California, United States of America, **5** Global Alliance for Livestock and Veterinary Medicine, Pentlands Science Park, Penicuik, Edinburgh, United Kingdom, **6** Glasgow Polyomics, Institute of Infection, Immunity and Inflammation, College of Medical, Veterinary and Life Sciences, University of Glasgow, Glasgow, United Kingdom, **7** Current address: Bioinformatics Group, Wageningen University, Wageningen, the Netherlands, **8** Institute of Parasitology, Biology Centre of the Czech Academy of Sciences, České Budějovice, Czech Republic

⊙ These authors contributed equally to this work.
* michael.barrett@glasgow.ac.uk

**Data Availability Statement:** Metabolomics data are available on MetaboLights (http://www.ebi.ac.uk/metabolights/) at accession numbers MTBLS1309 (T. congolense) and MTBLS1474 (T.

## Abstract

Livestock diseases caused by *Trypanosoma congolense*, *T. vivax* and *T. brucei*, collectively known as nagana, are responsible for billions of dollars in lost food production annually. There is an urgent need for novel therapeutics. Encouragingly, promising antitrypanosomal benzoxaboroles are under veterinary development. Here, we show that the most efficacious subclass of these compounds are prodrugs activated by trypanosome serine carboxypeptidases (CBPs). Drug-resistance to a development candidate, AN11736, emerged readily in *T. brucei*, due to partial deletion within the locus containing three tandem copies of the *CBP* genes. *T. congolense* parasites, which possess a larger array of related *CBPs*, also developed resistance to AN11736 through deletion within the locus. A genome-scale screen in *T. brucei* confirmed CBP loss-of-function as the primary mechanism of resistance and CRISPR-Cas9 editing proved that partial deletion within the locus was sufficient to confer resistance. CBP re-expression in either *T. brucei* or *T. congolense* AN11736-resistant lines restored drug-susceptibility. CBPs act by cleaving the benzoxaborole AN11736 to a carboxylic acid derivative, revealing a prodrug activation mechanism. Loss of CBP activity results in massive reduction in net uptake of AN11736, indicating that entry is facilitated by the concentration gradient created by prodrug metabolism.

## Author summary

AN11736 is a member of the benzoxaborole class identified as a development candidate for animal African trypanosomiasis, a deadly livestock disease with huge economic

brucei). Genome data are available on European Nucleotide Archive (ENA) and Sequence Read Archive (SRA) at accession number PRJEB34627.

**Funding:** This work was funded by a BBSRC grant (BB/S001034/1) and Global Alliance for Livestock Veterinary Medicine (GALVmed) with funding from BMGF and UKAID grant (OPP-1093639) to L.J.M. and M.P.B., a Wellcome Trust Senior Investigator Award to D.H. (100320/Z/ 12/Z), a Wellcome Trust Centre Award (203134/Z/ 16/Z) to Dundee and a core grant to the Wellcome Centre for Integrative Parasitology (104111/Z/14/ Z). The funders had no role in study design, data collection and analysis, decision to publish, or preparation of the manuscript.

**Competing interests:** The authors have declared that no competing interests exist.

impact. As part of its early evaluation phase, we set to unravel the risk and mode of resistance to this new trypanocide. We discovered that AN11736 behaves as a prodrug that, once inside trypanosomes, is cleaved by the activity of specific serine carboxypeptidases. AN11736-resistant *Trypanosoma brucei* and *T. congolense* had deletions within the serine carboxypeptidase gene array, resulting in their being unable to efficiently process the parent drug. Other benzoxaboroles with a similar sub-structure are also substrates for the serine carboxypeptidases, hence our findings assume great importance in considering the future development and deployment of this class of compounds.

## Introduction

Development of new drugs for infectious diseases has taken new urgency in recent years given the emergence and spread of antimicrobial resistance [1] that threatens global health. Resource-poor countries, where the infectious disease burden is highest, are most at risk. Drug resistant veterinary pathogens seriously compromise global food security. Animal African trypanosomiasis (AAT or nagana) affects millions of domestic animals each year [2] causing billions of dollars' worth of lost productivity in a part of the world where food scarcity impacts the population heavily. Given the rise of resistance to existing trypanocides [3], the Global Alliance for Livestock Veterinary Medicines has developed a programme to seek new drugs for AAT (https://www.galvmed.org/livestock-and-diseases/livestock-diseases/animal-african-trypanosomosis/). The leading class are the benzoxaboroles [4–7], boron-containing compounds that display versatile therapeutic potential against various infectious diseases [8]. Acoziborole is undergoing Phase II/III clinical trials for human African trypanosomiasis (HAT) [9, 10], a neglected tropical disease with unmet medical needs [11]. Acoziborole may play a key role in the HAT elimination programme [12], being active against both bloodstream and CNS involved stages of the disease after a single, oral dose [9].

More recently, another benzoxaborole, AN11736, was identified as a potential development candidate for AAT [13]. AN11736 cures cattle of both *Trypanosoma congolense* and *Trypanosoma vivax* infection as a single 10 mg/kg dose [13]. Compared to other benzoxaboroles, AN11736 is extremely potent against trypanosomes, killing at doses two to three orders of magnitude lower than that of the earlier AAT benzoxaborole candidate AN7973 [14] and of acoziborole [15], respectively.

As novel chemical entities, the benzoxaboroles are unlikely to display cross-resistance with current trypanocides. However, characterisation of mode of action and resistance mechanisms of these compounds are only now starting to emerge. Acoziborole resistance was initially associated with multiple genetic changes [16]. Subsequently, acoziborole, AN11736 and AN7973 were shown to target the Cleavage and Polyadenylation Specificity Factor 3 (CPSF3) which, when over-expressed, reduced drug sensitivity [15]. CPSF3 has also been identified as a target for benzoxaboroles in two apicomplexan parasites [17, 18], although in highly divergent organisms, i.e. bacteria and fungi, other targets have been proposed, including tRNA synthases [19, 20] and beta-lactamase [21]. In *Trypanosoma brucei* treated with acoziborole, metabolomics experiments revealed a profound change in methionine metabolism [22], that may relate to RNA processing defects, given multi-methylation of the spliced leader sequence used for *trans*-splicing by trypanosomatids [23]. Some processing mechanisms of particular benzoxaborole molecules by parasites have been identified. A trypanocidal benzoxaborole, of the amino-methyl subclass, was shown to be subject to two-step metabolic processing, involving a primary conversion by an amine oxidase in host serum to an aldehyde, that is further

metabolised to a carboxylate *via T. brucei* aldehyde dehydrogenase [24]. More recently, the benzoxaborole AN13762 was found to be intracellularly hydrolysed in *Plasmodium falciparum* by a lysophospolipase homologue, whose loss of function was linked to resistance [25].

Here we report on the risk and mode of resistance to AN11736 in animal trypanosomes. Our results indicate that AN11736 acts as a prodrug that, once inside trypanosomes, is cleaved by specific serine carboxypeptidases, thus creating a concentration gradient resulting in more parent drug entering the trypanosome cell. Loss or reduction of this enzymatic activity renders trypanosomes highly resistant to AN11736 and to related benzoxaboroles containing a common peptidic linker between the boron head group and a secondary moiety.

## Results

### Selection of resistance to AN11736 in *T. brucei* and *T. congolense*

Resistance to AN11736 was selected in *T. brucei* and *T. congolense* by continuous culture in escalating doses of drug. *T. brucei* able to grow in the presence of 9–18 nM of AN11736 were obtained after ~30 days of culture (Fig 1A). Resistant clones (one each of two independent resistant lines) were around 200-fold (TbOX$^R$_A) and >300-fold (TbOX$^R$_C) less sensitive to AN11736 as compared to parent cells.

For *T. congolense*, *in vitro* cultivation for more than eight months, in the presence of increasing concentrations of the compound, was required to reach high-level resistance. One individual clone from each of two different resistant lines, able to grow in the presence of 24–50 nM of the benzoxaborole, was chosen for subsequent studies. The sensitivity of these parasites to AN11736 decreased by >50-fold (clone TcoOX$^R$_B) to nearly 200-fold (clone TcoOX$^R$_C) as compared to the parent line (Fig 1B).

The *T. congolense* resistant clones grew only slightly slower than the parent line, but the growth rate was further reduced when the trypanosomes were cultured in the presence of AN11736 (S1A Fig). The resistance phenotype of these clones was stable after three months of growth in the absence of AN11736 (S1B Fig). Parent but not AN11736-resistant *T. congolense* parasites were cleared in *in vivo* mouse infections upon treatment with 5 mg/kg AN11736 (S1C Fig).

The *T. congolense* AN11736-resistant parasites did not demonstrate cross-resistance to drugs currently licensed for AAT, nor to other trypanocides used for HAT (S1 Table). Notably, no cross-resistance to the clinical candidate acoziborole was found (S1 Table).

Further cross-resistance analysis in *T. congolense* using a diverse array of benzoxaboroles revealed that AN11736-resistant trypanosomes were cross-resistant to compounds with a peptide-bond linker containing a valinate-amide motif, whereas these parasites showed no cross-resistance to benzoxaboroles without the linker (Fig 1C; see S2 Table for all data). These results agree with the absence of cross-resistance with acoziborole, which lacks the linker. Similar findings were obtained for *T. brucei* AN11736-resistant parasites when tested against various trypanocides and a selection of the same benzoxaboroles array (S3 Table).

### Resistant trypanosomes share genetic changes in a tandem array of serine carboxypeptidase (CBP) genes

Genome sequencing of the two *T. brucei* resistant clones TbOX$^R$_A and TbOX$^R$_C revealed a notable reduction of read depth in a region on chromosome 10 where a tandem repeat of the three serine peptidases *TbCBP1A*, *TbCBP1B* and *TbCBP1C* (Tb927.10.1030–1050 respectively) is present (Fig 2A). Evidently, a deletion of one or two *TbCBPs* alleles had occurred in TbOX$^R$_A and TbOX$^R$_C, which also exists in a region of apparent loss of heterozygosity.

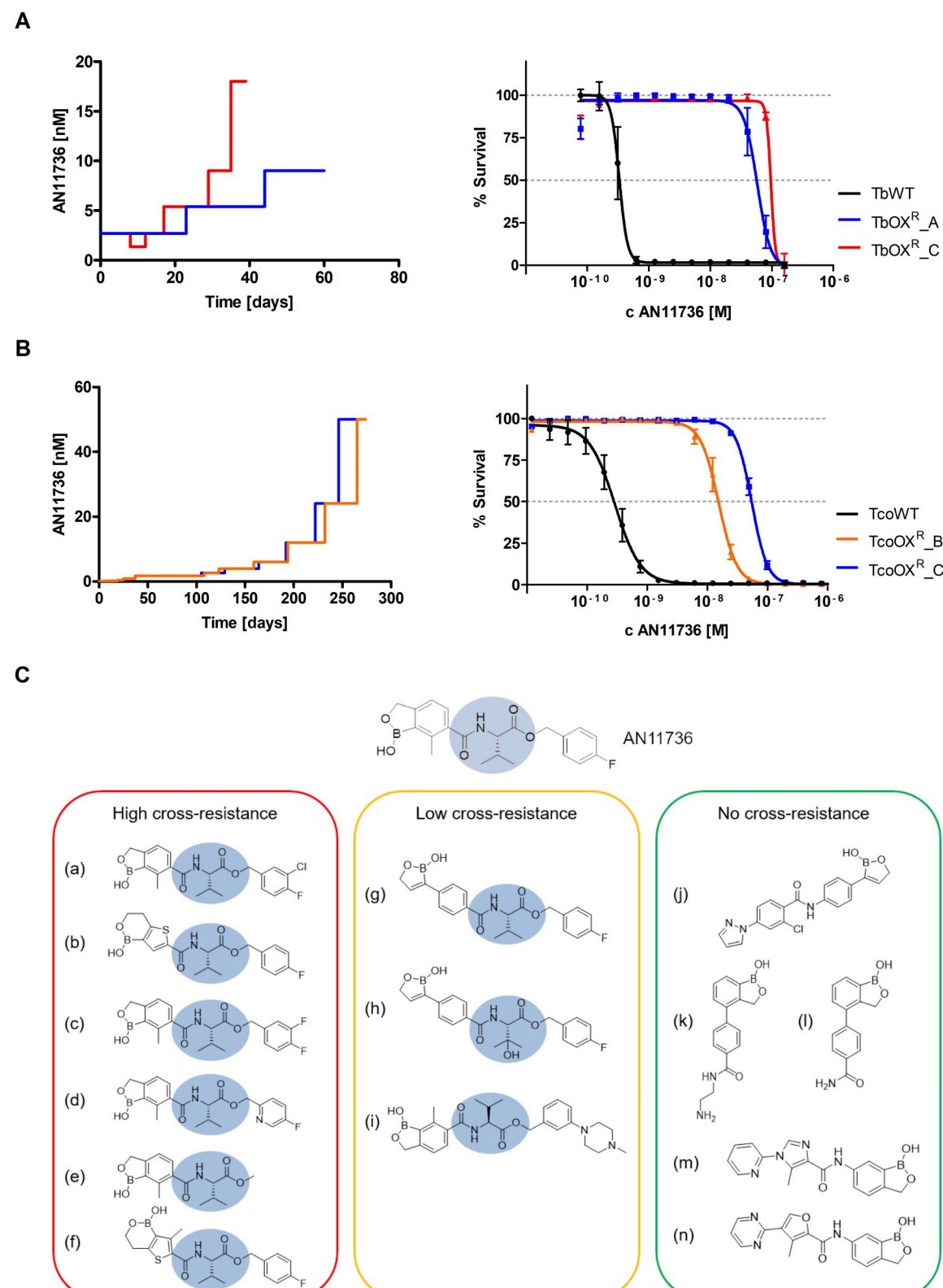

**Fig 1. AN11736 resistance selection in *T. brucei* and *T. congolense* and cross-resistance phenotype of the *T. congolense* AN11736-resistant clones, revealing a common chemical feature.** (A) *In vitro* selection of resistance to AN11736 in independent *T. brucei* lines TbOX^R_A and TbOX^R_C (left) and resistance levels of two resistant clones obtained from lines TbOX^R_A (EC$_{50}$ 58 nM)

and line TbOX$^R$_C (EC$_{50}$ 97 nM) as compared to the wild type line (TbWT, 0.3 nM) (right). (B) Stepwise *in vitro* selection of resistance to AN11736 in independent *T. congolense* lines TcoOX$^R$_B and TcoOX$^R$_C (left) and resistance levels of two resistant clones obtained from lines TcoOX$^R$_B (EC$_{50}$ 15 nM) and line TcoOX$^R$_C (EC$_{50}$ 54 nM) as compared to the wild type parent line TcoWT (EC$_{50}$ 0.3 nM) (right). (C) Cross-resistance of the *T. congolense* AN11736-resistant clones to other benzoxaboroles revealed the presence of a peptide-bond linker (highlighted in blue) in the highly cross-resistant compounds (>20-fold), whereas the same chemical feature was absent in non cross-resistant compounds (< 2-fold). See S2 Table for full data. Values in (A), (B) (right panels) represent means ± SEM of $n \geq 4$ (A) or $n = 3$ (B) independent biological replicates, with data in (B) each generated from two technical replicates.

Genome sequencing of the *T. congolense* resistant clones TcoOX$^R$_B and TcoOX$^R$_C and the parent lines (TcoWT, cultured for a limited number of passages, and TcoWT_HP, high passage, maintained in culture for the same time required for drug resistance selection), revealed reduced read coverage across the syntenic region of chromosome 10. This region comprises of nine annotated *TbCBP1* paralogues in the *T. congolense* IL3000 reference genome (here referred to as *TcoCBP1A-I*) (Fig 2B), although the read counts across the *T. congolense* CBP locus suggest a different arrangement of paralogues in our experimental strain relative to the reference genome assembly. We observed a pronounced drop in read depth at the 5' end of *TcoCBP1A* and rise across the gene *TcoCBP1I*, indicating a loss of several CBPs in resistant cells. The high sequence conservation between the genes complicates analysis. TcoCBP1A and TcoCBP1I are the most divergent from the other seven TcoCBPs (S2 Fig). Five genes (*TcoCBP1C*, *TcoCBP1D*, *TcoCBP1F-H*) appeared to have the least mappable reads and hence were most likely deleted in both lines. *TcoCBP1A*, *TcoCBP1B* and *TcoCBP1I* did not appear to be affected. We were not able to identify homozygous SNPs in *TbCPSF3* or *TcoCPSF3* that could explain the resistance phenotype.

## Knockdown of serine carboxypeptidases causes AN11736 resistance in *T. brucei*

RNA interference (RNAi) target sequencing (RIT-seq) provides a means to identify genes whose knockdown promotes drug resistance [26]. Selecting a library of *T. brucei* cells containing RNAi-inducing constructs covering the whole genome in the presence of AN11736 also identified CBPs knockdown as the dominant 'hit' conferring resistance (Fig 3A). Moreover, silencing the expression of the *TbCBP1* genes by targeted RNAi confirmed their importance for sensitivity to AN11736, as tetracycline induction of RNAi in these trypanosomes increased the EC$_{50}$ ~25-fold (Fig 3B).

Disruption of *TbCBP1A-C* (Tb927.10.1030–1050) function by CRISPR-Cas9 gene editing [27] corroborated these results. Cas9 programmed to target *TbCBP1A-C* was induced for 24 h and then cells were selected with AN11736 using two independent Cas9/sgRNA$^{CBP1}$ clones. The growth profiles indicated robust drug-resistance with induction of *TbCBP1A-C* editing that was not observed in wild type control cells (Fig 3C). A PCR-based assay confirmed that the *CBP1* locus was disrupted in both independent edited clones (Fig 3D, upper panel). Consistent with repair by single-strand annealing [28], sequencing of the products revealed recombination within blocks of identity in the 1050 (*TbCBP1C*) and 1030 (*TbCBP1A*) genes (Fig 3D, lower panel; see S3 Fig for clone 1 sequence). Notably, these cells retained a chimeric copy of *CBP1A* and *CBP1C*, suggesting that the chimeric protein fails to sensitise parasites to the drug. This may also be the case for the paralogues retained by the resistant strains described above that emerged following drug selection (Fig 2). Assessment of clones' sensitivity to AN11736 revealed a near 200-fold increase in EC$_{50}$ for clone 1 and a 300-fold increase in EC$_{50}$ for clone 2 (Fig 3E). Thus, CRISPR-Cas9 editing confirmed the role of these serine carboxypeptidases in sensitivity to AN11736.

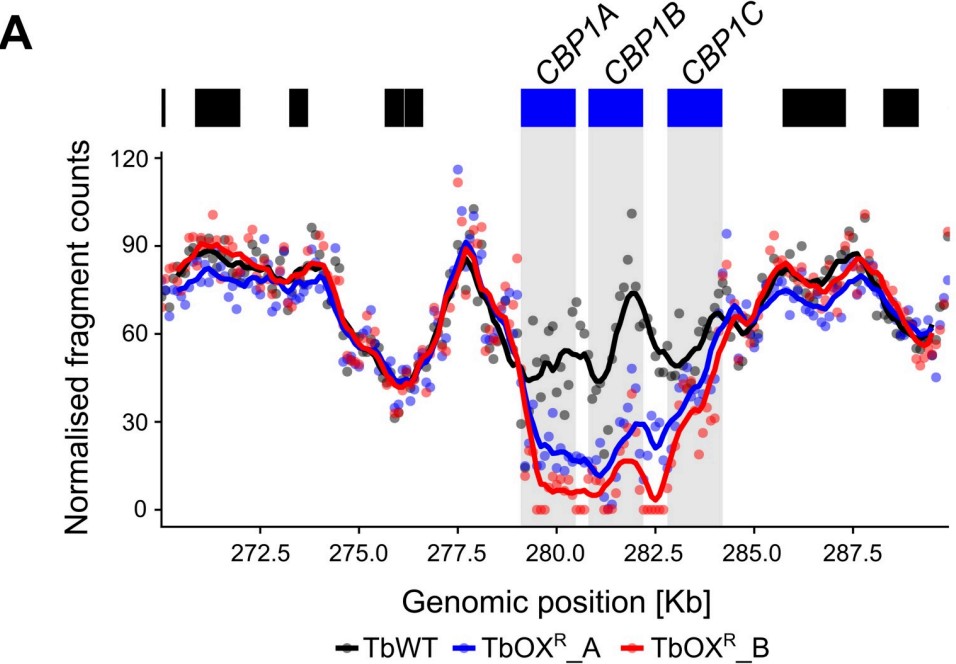

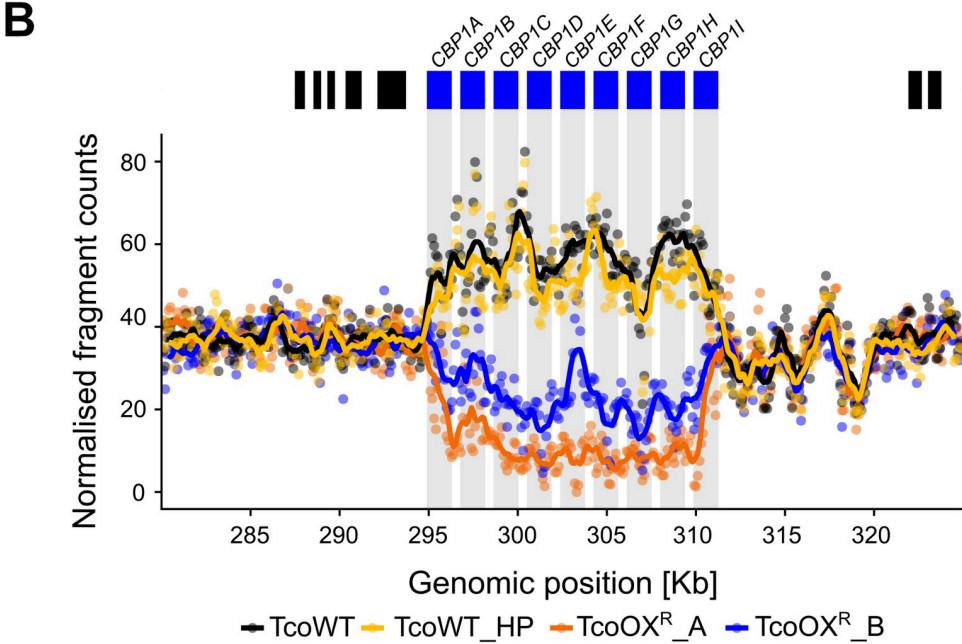

**Fig 2. Serine carboxypeptidases are deleted in *T. brucei* and *T. congolense* resistant to AN11736.** (A) Coverage of whole genome sequencing data for wild type (TbWT) and AN11736-resistant (TbOX^R_A, TbOX^R_B) *T. brucei* at the genomic locus of *CBP1* gene copies. The number of fragments mapping to 100 bp windows is shown as individual points, with a moving average of nine windows shown as a line plot. Coverage was normalised for depth of sequencing as described in Materials and Methods. The position of genes within this region is shown as blocks above the plot, with *CBP1* genes in blue and other genes in black. (B) Coverage plot as in (A) but with *T. congolense* wild type (both parent, TcoWT, and high passage, TcoWT_HP) and AN11736-resistant lines (TcoOX^R_A, TcoOX^R_B).

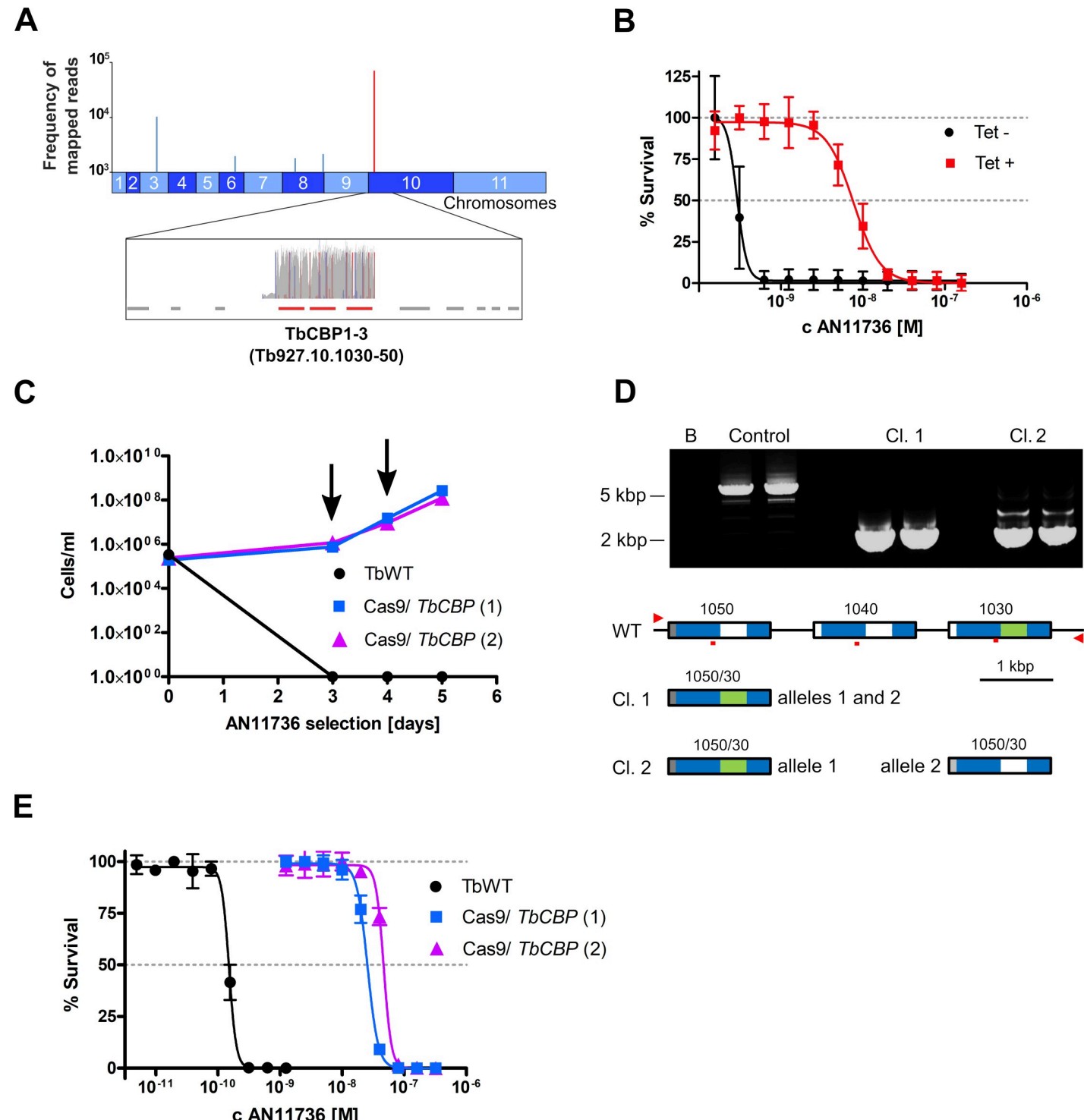

**Fig 3. Serine carboxypeptidase disruption confers resistance to AN11736.** (A) Genome-wide RIT-seq screen in *T. brucei* identified the *CBP1* genes on chromosome 10 as 'hits' for AN11736 resistance. In the lower panel, blue and red peaks are forward and reverse barcoded reads from individual RNAi target fragments; grey, all other reads. (B) Targeted *CBP1* RNAi knockdown of the three serine carboxypeptidases in *T. brucei* conferred resistance to AN11736 under tetracycline induction (Tet + EC$_{50}$ 7.7 nM, Tet- EC$_{50}$ 0.3 nM). (C) Cumulative cell growth in the presence of 10 nM AN11736 (arrows indicate addition of fresh drug) for uninduced *T. brucei* and two independent induced Cas9/sgRNA$^{CBP1A-C}$ clones, following 24 h of Cas9 induced editing in the latter case. (D) A PCR assay of the Cas9/sgRNA$^{CBP1A-C}$ clones revealed *TbCBP1* editing in two independent drug-resistant clones (upper panel, duplicate samples are shown); the locus from the uninduced line yielded a product of approximately 5.7 kbp, while both clones yielded a product of approximately 2 kbp. B (blank), no genomic DNA; control, uninduced line genomic DNA; predicted wild type *CBP1* fragment size, 5,696 bp. The maps in the lower panel indicate the edited loci, as determined by sequencing. The small red bars indicate the gRNA

target-sites; the red arrowheads indicate the primers used for the PCR-assay; blue indicates >99% identical regions among multiple paralogues; grey, unique to Tb927.10.1050; green, unique to Tb927.10.1030. (E) Dose-response curves for AN11736 of the two CRISPR-Cas9 edited clones analysed, both displaying a drug-resistant phenotype: when CBP1 function was disrupted by Cas9 editing, *T. brucei* became, on average, 250-fold more resistant to AN11736. Data in (B), (E) represent means ± SD of *n* = 3 independent biological replicates.

## Re-expression of serine carboxypeptidases re-sensitises AN11736-resistant trypanosomes to the drug

Re-expression of a functional copy of TbCBP1B re-sensitised *T. brucei* TbOX$^R$_A to AN11736 (Fig 4A). Trypanosomes retain the catalytic triad Ser-Asp-His of carboxypeptidases, identified by alignment with other serine carboxypeptidases belonging to the S10 family, well characterised in yeast [29, 30] and the other trypanosomatid *T. cruzi* [31] (S4 Fig). Disruption of the *T. brucei* catalytic triad in the active site of TbCBP1B, by substituting the nucleophilic S179 with a hydrophobic alanine, failed to re-sensitise the cells to the drug using the same approach (Fig 4A). Re-expression of TcoCBP1A and TcoCBP1H in TcoOX$^R$_C also partially restored sensitivity to AN11736 (Fig 4B). Heterologous re-expression of TbCBP1B in TcoOX$^R$_C (Fig 4C) partially re-sensitised the parasites to AN11736. A similar effect was obtained when re-expressing the only predicted CBP1 serine carboxypeptidase annotated in the *T. vivax* genome in TbOX$^R$_A (Fig 4D). Heterologous expression of TcoCBP1H but not TcoCBP1A in TbOX$^R$_A restored sensitivity (Fig 4E).

These results prove that serine carboxypeptidases sensitise different *Trypanosoma* species to the benzoxaborole AN11736 and that loss of these genes renders the parasites less sensitive to the compound.

## Trypanosome serine carboxypeptidases cleave AN11736 to a carboxylate derivative

Mass spectrometry analysis of *T. brucei* wild type and resistant parasites treated for 6 h with a high dose of AN11736 (0.9 μM, > 1,000-fold EC$_{50}$) showed the compound was present in the parent and both resistant lines, indicating there was no defect in uptake (Fig 5A). Further analysis revealed a compound fragment that was present in the wild type, but not the resistant lines (m/z 292.1347, retention time 10 minutes) (Fig 5B). This fragment had a boron isotope distribution (S5 Fig) and a predicted formula of $C_{14}H_{19}O_5NB$. A simulation of the isotopic distribution of $C_{14}H_{19}O_5NB$ matched the pattern for m/z 292.1347 (S5 Fig), further corroborating the identification of this metabolite as a fragment of AN11736, generated by cleavage within the linker at the level of the ester bond.

In *T. congolense* treated for 6 h with 2 μM AN11736 we could detect AN11736 in parent wild type, both resistant lines and the CBP1H-complemented line (Fig 5C). In *T. congolense* the $C_{14}H_{19}O_5NB$ fragment was identified as a large peak in the wild type line but at substantially lower levels in the resistant lines TcoOX$^R$_C and TcoOX$^R$_B (Fig 5D). The CBP1H-complemented resistant line TcoOX$^R$_C regained higher levels of the product, proportional to regaining sensitivity to the drug.

Taken together, these data reveal that AN11736 acts as a prodrug that, once inside trypanosomes, is cleaved by serine carboxypeptidases at the ester bond to give a carboxylate derivative (m/z 292.1347, later synthesized under the code name AN14667). In resistant trypanosomes, where genes encoding for serine carboxypeptidases have been deleted or disrupted, this activation does not occur, or does so with substantially reduced efficiency.

When tested against trypanosomes, the carboxylate derivative AN14667 showed much reduced activity compared to AN11736 (~15,000-fold less active against *T. brucei* wild type and ~800-fold less active against *T. congolense* wild type), most likely explained by the charged carboxylate derivative poorly traversing the parasite membrane (S6 Fig).

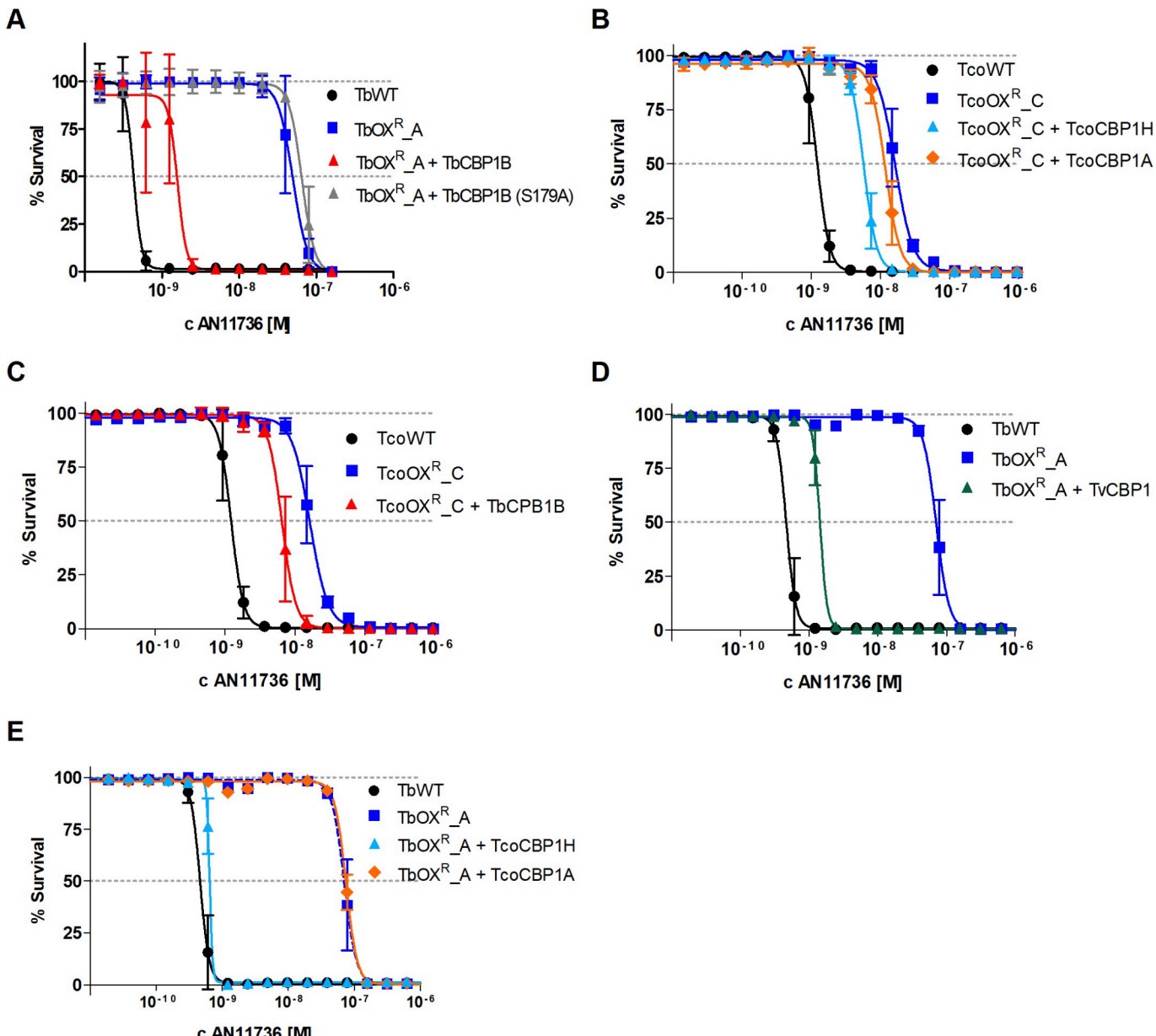

**Fig 4. Re-expression and heterologous expression of serine carboxypeptidases re-sensitise AN11736-resistant trypanosomes.** (A) Re-expression of TbCBP1B (Tb927.10.1040) in resistant line TbOX$^R$_A partially re-established sensitivity to AN11736 (EC$_{50}$ 1.6 nM), except when serine 179 was replaced with alanine in the catalytic triad (TbOX$^R$_A + TbCBP1B (S179A), EC$_{50}$ 65 nM). (B) Add-back of TcoCBP1H and, to a lesser extent, TcoCBP1A partially restored sensitivity to AN11736 in the resistant line TcoOX$^R$_C (EC$_{50}$ 5.7 nM and EC$_{50}$ 11.7 nM respectively, TcoOX$^R$_C EC$_{50}$ 16.3 nM). (C) Heterologous expression of TbCBP1B (EC$_{50}$ 6.5 nM) partially restored sensitivity to AN11736 in resistant TcoOX$^R$_C line (EC$_{50}$ 16.3 nM) compared to TcoWT (EC$_{50}$ 1.2 nM). (D) Expression of TvCBP1 (EC$_{50}$ 1.4 nM) partially restored sensitivity to AN11736 in resistant TbOX$^R$_A line (EC$_{50}$ 70.7 nM) compared to TbWT (0.46 nM). (E) Expression of TcoCPB1H (EC$_{50}$ 0.65 nM) but not TcoCBP1A (EC$_{50}$ 75.1 nM) restored sensitivity to AN11736 in resistant TbOX$^R$_A line (EC$_{50}$ 70.7 nM) compared to TbWT (0.46 nM). Data represent means ± SD of *n* = 3 independent biological replicates.

## AN11736 metabolism causes accumulation of AN14667 and sustains further internalization of the parent compound in sensitive cells

Absolute quantification of AN11736 and its carboxylate metabolite AN14667 by UPLC-MS/MS in wild type and resistant parasites substantiated these findings (Fig 5E and 5F). Over a

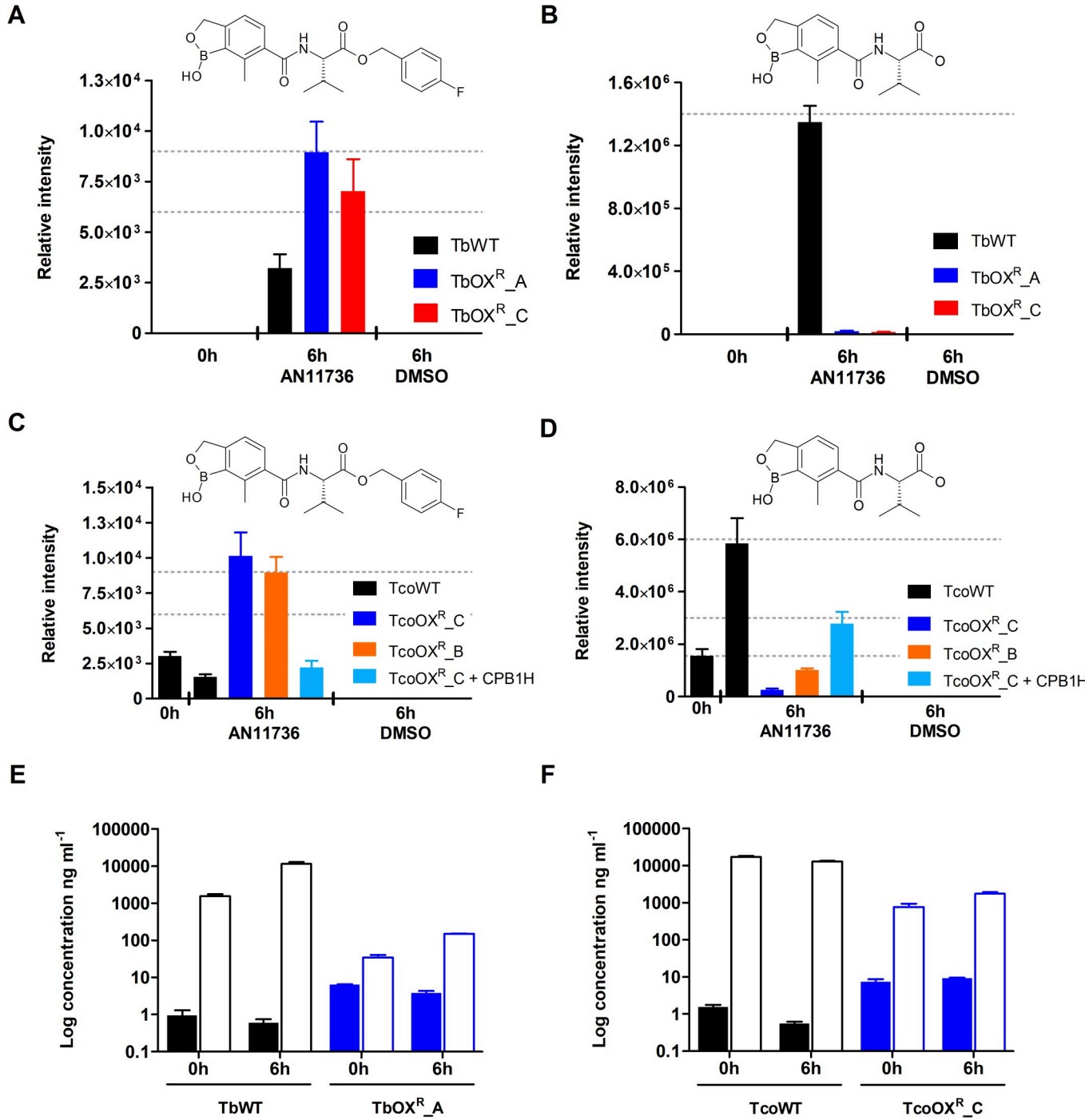

**Fig 5. AN11736 is cleaved at the ester bond to a carboxylate derivative, which accumulates at high levels in wild type, but not resistant *T. brucei* or *T. congolense*.** (A) LC-MS analysis revealed presence of AN11736 in TbWT and resistant lines TbOX$^R$_A and TbOX$^R$_C after 6 h of incubation. (B) A peak of m/z 292.1347 was detected in positive mode in TbWT cells after 6 h of incubation but was barely detectable in resistant cell lines TbOX$^R$_A and TbOX$^R$_C. The peak was identified as the AN11736 carboxylate derivative AN14667, whose chemical structure is shown above the graph. (C) AN11736 was detected in TcoWT, the resistant lines TcoOX$^R$_C, TcoOX$^R$_B and the CBP1H-complemented TcoOX$^R$_C line after 6 h of incubation. (D) The peak of m/z 292.1347 was detected in positive mode in TcoWT cells after 6 h of incubation and at around half the intensity for re-sensitised CBP1H add-back line TcoOX$^R$_C + CPB1H, while the peak was detected at very low levels in resistant cell lines TcoOX$^R$_C and TcoOX$^R$_B, with intensities even lower than that measured for TcoWT at 0 h. (E) Mass spectrometry quantification of AN11736 (full bars) and the metabolite AN14667 (empty bars) in *T. brucei* wild type (black) and AN11736 resistant line TbOX$^R$_A (blue) over a period of 6 h. (F) as in (E) but for *T. congolense* wild type and AN11736 resistant line TcoOX$^R$_C. Data represent means ± SD of $n = 3$ (A-D) or $n = 2$ (E, F) independent biological replicates.

period of 6 h, relatively unchanged intracellular amounts of AN11736 (1 ng ml$^{-1}$ at 0 h–essentially cells centrifuged as soon as possible after addition of drug, and 0.6 ng ml$^{-1}$ at 6 h) were measured in TbWT, while in these cells levels of AN14667 were already 1,600-fold higher at time 0 h and further increased at 6 h (1,560 ng ml$^{-1}$ at 0 h and 11,436 ng ml$^{-1}$ at 6 h), indicating a very fast processing of the parent compound. In the resistant line TbOX$^R$_A the opposite was observed: levels of AN11736 were higher at both timepoints (6.4 ng ml$^{-1}$ at 0 h and 3.85 ng ml$^{-1}$ at 6 h), while AN14667 levels remained much lower (34.5 ng ml$^{-1}$ at 0 h and 150.5 ng ml$^{-1}$ at 6 h) than those found in TbWT (Fig 5E). Quantification of these metabolites in *T. congolense* corroborated the *T. brucei* data. Levels of AN11736 remained low and slightly decreased over time in TcoWT (1.6 ng ml$^{-1}$ at 0 h and 0.6 ng ml$^{-1}$ at 6 h), while during the same time levels of AN14667 were markedly higher (17,272 ng ml$^{-1}$ at 0 h and 12,814 ng ml$^{-1}$ at 6 h). In TcoOX$^R$_C, AN11736 was present at higher concentration (7.5 ng ml$^{-1}$ at 0 h and 9.2 ng ml$^{-1}$ at 6 h) than in the wild type line, while its metabolite levels remained lower than in the wild type (767 ng ml$^{-1}$ at 0 h and 1,757 ng ml$^{-1}$ at 6 h) (Fig 5F).

*T. brucei* over-expressing CPSF3 were less sensitive to both acoziborole and AN11736 [15] indicating that both drugs act by inhibiting this target. The differential activity between acoziborole and AN11736 would, therefore, appear to be related to the metabolism of the parent drug by CBP1, creating a more potent, charged derivative that is retained in the cell, as previously shown for another benzoxaborole by a distinct metabolic route [24]. Hence, AN11736 would enter cells down a concentration gradient that is perpetuated by drug metabolism with the cleaved derivative accumulating to concentrations much higher than the parent drug. As absolute quantification shows, the metabolite AN14667 reaches vastly higher levels than the parent compound, supporting the superior activity of AN11736 as compared to other benzoxaboroles that do not undergo enzymatic activation, relating to a far greater intracellular accumulation of active compound inside parasites.

## Discussion

The benzoxaborole class of compounds has produced multiple clinical development candidates against a range of conditions, including infectious disease [4–7]. Acoziborole, for example, is in clinical trials for human African trypanosomiasis [9, 10] and AN11736 is a member of a highly potent benzoxaborole subclass currently under consideration for treatment of AAT [13]. Recently, the conserved splicing factor CPSF3 was proposed as the major cellular target for benzoxaboroles in trypanosomes [14, 15], supported by an earlier study that revealed, among other changes, amplification of the CPSF3, selected during induction of resistance in *T. brucei* [16]. AN11736 and a series of related compounds have potency against trypanosomes that exceed that of acoziborole by two to three orders of magnitude. Given requirements of high potency (to keep costs as low as possible for use in cattle), these compounds have received particular interest.

As part of the development process, understanding the risk and mechanisms of resistance is crucial. Here we reveal that the high potency of AN11736 is related to prodrug processing: once the compound has entered trypanosomes it is cleaved by serine carboxypeptidase(s) to a carboxylate product trapped within the cell. This enables accumulation of the benzoxaborole to greatly exceed that where no prodrug conversion and entrapment occurs. The same process, however, leads to a less desirable situation where selection of resistance becomes possible due to loss of enzyme(s) involved in prodrug processing.

Resistance to AN11736 occurs by disruption of expression of serine carboxypeptidase (CBP) genes, which results in diminished AN11736 cleavage. This mechanism of prodrug activation appears analogous to one recently observed in *P. falciparum*, where benzoxaborole AN13762 is cleaved by esterase activity, whose loss confers resistance to the compound [25].

The CBPs have been characterised in the trypanosomatid *T. cruzi* [31]. In this parasite, the C group serine peptidases in the S10 serine peptidase family proteolytically cleave at C-termini at acidic pHs. This cleavage happens in lysosomes [31], where the enzymes also have esterase and deamidase activities [32, 33]. In *T. cruzi*, activation of serine peptidases may be achieved through cleavage of a pro-domain by cruzipain [34]. *T. brucei* serine carboxypeptidases also have a pro-domain, but it is not known whether brucipain, the cruzipain homologue, is required for *T. brucei* serine peptidase activation. It is possible that mutations in brucipain would result in a secondary mechanism of resistance, although this was not observed in our analysis. It is probable that the *T. brucei* and *T. congolense* CBP serine carboxypeptidases play similar roles to that in *T. cruzi* where its lysosomal localisation and multiple hydrolytic capabilities [32, 33] likely play a generic role in macromolecule turnover. We have not ascertained whether the genes are essential in procyclic form parasites which are resident in the tsetse fly. It will be important to understand this in the future, since a fitness cost in this lifecycle stage would hinder the transmission of parasites that develop resistance via this route in the mammalian bloodstream.

Gene deletions in the serine carboxypeptidase arrays of both *T. congolense* and *T. brucei* could clearly be linked to resistance to AN11736. Due to the high degree of sequence homology of the CBPs in both *T. brucei* and *T. congolense* we were unable to identify the precise CBP gene deletion(s). However, resistance to AN11736 occurred relatively quickly in *T. brucei in vitro*, while for *T. congolense*, which possesses a larger array of CBP genes, resistance took longer to emerge. Importantly, *T. congolense* AN11736-resistant parasites retained infectivity and resistance phenotype in mice.

The ~200-fold level of resistance obtained for both *T. brucei* and *T. congolense* indicates that potency of the otherwise hyper-potent AN11736 would be similar to that of many other benzoxaboroles, including acoziborole (500 nM against *T. congolense* and 270 nM against *T. brucei*) [22] in absence of drug processing. This suggests that maintaining a concentration in animals that would still kill, even if drug activation were lost, could be possible, *albeit* using much higher doses of AN11736, which might compromise economic development.

Conversely, a similar cleavage of AN11736 could occur through peptidases present in the blood of treated animals, hence affecting pharmacokinetics. This possibility could reduce the amount of parent compound in circulation, an occurrence particularly important in view of potential prophylactic applications. Our data suggest that the pre-processed compound is of much reduced activity, presumably as the charged derivative is membrane impermeant, consistent with the sink effect resulting from intracellular generation of a carboxylate product.

Experiments with heterologous expression of serine carboxypeptidases suggest that benzoxaborole activation by ester cleavage identified for *T. congolense* and *T. brucei* is most likely shared with other trypanosomes, or at least with the major veterinary species *T. vivax*. In *T. vivax* the CBP locus in the Y486 strain reference genome consists of a single gene. Whether this would make resistance easier to acquire, or conversely more difficult, given the lack of redundancy, should be investigated once an *in vitro* culturing system for *T. vivax* has been developed.

As well as elucidating the resistance mechanism to a class of potent benzoxaboroles, the discovery of a particular moiety that is specifically cleaved by trypanosomal carboxypeptidases offers the potential to exploit that linker to create novel prodrugs with targeted activity against trypanosomatids, i.e. drugs that may not have the ability to traverse membranes could be linked, via the CBP cleavable bridge, to hydrophobic moieties enabling diffusion into cells where they would be cleaved to release the specific inhibitor. A risk of resistance to such compounds emerging through mutation to the CBP genes, though, could limit such a use.

## Materials and methods

### Ethics statement

The mouse experiment was carried out in accordance with the Animals (Scientific Procedures) Act 1986 and the University of Glasgow care and maintenance guidelines. All animal protocols and procedures were approved by the Home Office of the UK government and the University of Glasgow Ethics Committee. Work was covered by Home Office Project Licence 60/4442.

### Trypanosome culture

Bloodstream form (BSF) *T. b. brucei* strain Lister 427 was cultured at 37˚C in a humidified, 5% $CO_2$ environment in HMI-11 (Gibco), supplemented with 10% FBS (Gibco). BSF *T. congolense* strain IL3000 was cultured at 34˚C in a humidified, 5% $CO_2$ environment in TcBSF-3 [35], containing 20% commercial goat serum (Gibco) and 5% Serum Plus II (SAFC Biosciences). Red blood cell lysate was not present.

### Generation of oxaborole-resistant lines

*T. congolense* and *T. brucei* parasites were selected for resistance to AN11736 by subculturing cells *in vitro* in the continuous presence of increasing concentrations of the compound. Multiple, independent cell lines were selected in parallel. Resistant lines were cloned by limiting dilution.

### Trypanocidal activity

*In vitro* trypanocidal activity was measured using the Alamar Blue method as previously described [36]. *T. brucei* were seeded at $2 \times 10^4$ cells ml$^{-1}$ and *T. congolense* at $2.5 \times 10^5$ cell ml$^{-1}$ and $EC_{50}$ values determined after a total drug incubation time of 72 h. All experiments were carried out in duplicate and on at least three independent occasions unless stated otherwise. Although the assay measures metabolic conversion of resazurin to resorufin reagent and this can also be hindered by trypanostatic compounds we believe in this case it is a true surrogate for cell death hence we labelled the y-axis as percentage parasite survival in Fig 1.

### *In vivo* virulence of benzoxaborole-resistant trypanosomes

200 μl of $4.5 \times 10^7$ *T. congolense* AN11736 resistant cells and $2.8 \times 10^7$ wild type in fresh TcBSF-3 were injected intravenously into immunocompromised NIH female mice (5 per group). Once high parasitaemia had developed (day 16), AN11736 (prepared as a suspension in 10% DMSO) was administered i.p.at a dose of 5 mg/Kg. Parasitaemia was monitored daily by tail blood examination and mice humanely culled when parasitaemia reached $10^8$ cells ml$^{-1}$.

### Whole genome analysis

DNA from *T. brucei* and *T. congolense* wild type and resistant clones was extracted using the NucleoSpin Tissue Kit (Macherey-Nagel). Sequencing of paired 75 bp reads was performed using the NextSeq 500 platform (Illumina). Libraries were prepared with 500 ng input gDNA using QIAseq FX DNA library kit (Qiagen) and fragments of 300 bp, including adaptors, were selected with Agencourt AMPure XP (BeckmanCoulter), according to manufacturer instructions. Reads were trimmed for quality and adaptor contamination using Trim Galore! v0.6.2 (Babraham Bioinformatics) and reads were aligned to either the *T. b. brucei* TREU 927 reference genome, release 43 (available from TriTrypDB, https://tritrypdb.org/tritrypdb/), or the *T. congolense* TcIL3000 2019 reference genome assembled from Pacific Biosciences sequencing data by N. Hall group (also available from TriTrypDB), using Bowtie2 v2.3.5 [37]. Alignment rates for all samples were 85% for

*T. brucei* and 98% for *T. congolense*. Reads were sorted and duplicates marked using SAMtools v1.9 [38] and Picard Tools v2.20.2 (http://broadinstitute.github.io/picard/). To determine depth of coverage, the number of fragments mapping to 100 bp windows along the genome was quantified using featureCounts v1.6.3 [39] with fragments mapping to multiple windows counted as 1/n (where n is the total number of windows to which a given fragment maps). For comparison of depth of coverage, the number of fragments mapping to each window was normalised for sequencing depth by dividing it by the ratio of number of reads aligned to the parent chromosome (chromosome 10) for that sample to the mean aligned reads for that chromosome across all samples.

## Genetic manipulation of *T. congolense* and *T. brucei*

For the re-expression of CBPs in *T. brucei* the amplified open reading frame sequences were cloned into the pRM481 vector [40] using *Xba*I and *Bam*HI (for primers list see S4 Table). Due to high sequence homology between the *TbCBP* genes the Tb427.10.1040 ORF was synthesized (BaseClear) and used as template for the PCR. The S179-encoding codon was modified in the above-described pRM481-derived vector containing wild type *TbCBP1B* by using Q5 Site-Directed Mutagenesis (SDM) Kit (NEB). Primers designed for this purpose were TGTTGGGGAAgcCTACGGTGGC and ACAAAGAAGTCGTTTTCAC.

A plasmid was generated that targets the tubulin locus of *T. congolense* and transcribes blasticidin S deaminase (BSD) and the C-terminal 6×HA tagged trypanosomal CBP described herein. *T. congolense* 5' and 3'-tubulin and actin intergenic sequences were amplified by PCR with Q5-Polymerase (NEB) from genomic DNA. Blasticidin S deaminase (*BSD*) was amplified from a plasmid pGL2271 (for primers list see S4 Table). Vector pRM481 was digested with *Asc*I and the plasmid backbone was purified and used for Gibson assembly (NEB) with PCR products. In the resulting vector the *CBP* genes were flanked upstream by actin intergenic sequence and *BSD*. Upstream and downstream of the *BSD* gene were the 5' and 3' tubulin intergenic sequences that allowed integration into the tubulin locus upon *Asc*I digest. This vector was further modified to accept the *CBP* genes: at 3' of *Bam*HI site a *Hpa*I site was introduced and the *Cla*I site, separating the gene (*GFP*) and the actin intergenic region, was exchanged with a *Sal*I site by SDM (for primers list see S4 Table).

Plasmid pRM481 encoding for TcoCBP1A was modified by SDM. A *Hpa*I site was introduced 3' of the 6×HA tag and the *Xma*I site. Then a PCR was performed to amplify the open reading frame and introduce a *Sal*I site 5' of the *TcoCBP1A* ATG and a primer that bound in the *T. brucei* 3' tubulin intergenic region (AAACCTACACATGGTGCGACG). *TcoCBP1A* was inserted into the pRM481 derivative with *Sal*I and *Hpa*I. Further gene exchanges were done by amplifying the ORF and inserting into the *Bam*HI and *Sal*I sites.

10 μg of *Asc*I-digested plasmid DNA were transfected into $3 \times 10^7$ *T. congolense* (TcoOX$^R$_C) and *T. brucei* (TbOX$^R$_A) cells as described [41]. *T. congolense* and *T. brucei* clones were selected with 0.4 μg ml$^{-1}$ blasticidin (InvivoGen) in TcBSF-3 and 5 μg ml$^{-1}$ blasticidin in HMI-11, respectively. Expression was verified by detection of 6×HA tagged protein and EF1α as loading control on the LiCor's Odyssey Imaging system.

Putative catalytic serine of TbCBP1B (S179) was identified by aligning its ORF sequence with a series of S10 serine peptidase ORF sequences downloaded from MEROPS Peptidase Database [42] including Carboxypeptidase Y from yeast, whose catalytic triad is well characterized [29, 30]. Alignment was conducted in CLC Genomics Workbench.

## RNAi screen

Determinants of AN11736 resistance were identified using an RNAi library screen as previously described [26]. Briefly, cultures from the screen were split and supplemented with fresh

AN11736 as required and DNA was extracted from drug-resistant cells. RNAi target fragments were amplified by PCR using the LIB2f and LIB2r primers. The products were then subjected to high-throughput RIT-seq. Sequencing was carried out on an Illumina HiSeq platform at BGI (Beijing Genomics Institute). Reads were mapped to the *T. brucei* 927 reference genome (v9.0, tritrypdb.org) with Bowtie2 using the parameter: very- sensitive-local-phred33. The generated alignment files were manipulated with SAMtools and a custom script was used to identify reads with barcodes (GCCTCGCGA) [43]. Total and bar-coded reads were then quantified using the Artemis genome browser [44].

## Targeted RNAi of CBP locus

A single RNAi construct targeting all three *TbCBPs* in a common region of their DNA sequence was produced using plasmid pGL2084 as a backbone, and the resulting vector was transfected into *T. brucei* 2T1 BSF as previously described [45]. Primers to amplify the RNAi target sequence included *AttB* gateway flanks: Fw: *GGGGACAAGTTTGTACAAAAAAGCAG GCT*CGTTAATCAATGGAGCGGAT, Rev: *GGGGACCACTTTGTACAAGAAAGCTGGGT* GCTTTCCCCAACAACAAAGA. Genetically modified parasites were selected in HMI-11 complemented with 0.5 μg ml$^{-1}$ phleomycin (InvivoGen), and 2.5 μg ml$^{-1}$ hygromycin B (Calbiochem). RNAi induction was obtained with tetracycline (Sigma-Aldrich) at 1 μg ml$^{-1}$ 24 h before experiments.

## CRISPR-Cas9 gene editing

*T. brucei* Lister 427 parasites were grown and manipulated as described previously [27]. The Cas9 gRNA oligonucleotide pair comprised CBP1.G1 (**AGGG**CCTCTTGCAGGATTGGCT GT) and CBP1.G2 (**AAAC**ACAGCCAATCCTGCAAGAGG); the overhanging ends that facilitate cloning are in bold. These were annealed, ligated to *Bbs*I-digested pT7$^{sgRNA}$, confirmed by sequencing and introduced into 2T1$^{T7-Cas9}$ cells as described [27]. DNA analysis and drug-sensitivity analysis were carried out as described [27]. The primer pair used for the Cas9 PCR assay comprised FwCBP1PCRout (GTTACAACATAACCACCGCGG) and RvCBP1PCRout (GGTGGAGTGGGCACAACCAC).

## Metabolomics analysis

*T. congolense* and *T. brucei* metabolites were extracted for untargeted metabolomics analysis following treatment with test compounds at $10 \times EC_{50}$ or with the DMSO vehicle control (below 1% v/v) for 6 hours. For each sample $1 \times 10^8$ cells were collected and their metabolism was quenched by rapidly cooling to 4˚C using a dry ice/ethanol bath. The cells were kept at 4˚C from hereon. After a wash in ice cold PBS, cells were resuspended in 200 μl of extraction solvent (Chloroform:Methanol:Water 1:3:1) and shaken at 4˚C for 1 h. Extracts were centrifuged at 17,000×g, 10 min, 4˚C and the supernatants collected and stored under argon at -80˚C until analysis by LC-MS. Four replicates of each sample were prepared. Samples were analysed on an Orbitrap Fusion mass spectrometer (Thermo Fisher Scientific) in both positive and negative modes (switching mode). Hydrophilic interaction liquid chromatography (HILIC) was carried out on a Dionex UltiMate 3000 RSLC system (Thermo Fisher Scientific) using a ZIC-pHILIC column (150 mm Å~ 4.6 mm, 5 μm column, Merck Sequant). HPLC mobile phase A was 20 mM ammonium carbonate in water and mobile phase B was 100% acetonitrile. The column was maintained at 30˚C and samples were eluted with a linear gradient from 80% B to 20% B over 24 minutes, followed by 8 minutes wash with 5% B and 8 minutes re-equilibration with 80% B, at the flow rate of 300 μl/minute. Orbitrap data were acquired as previously described [46]. Untargeted peak-picking and peak matching from raw LC-MS data

were obtained using XCMS and mzMatch respectively. Metabolite identification and relative quantitation was performed using IDEOM interface [46] and PIMP [47], by matching accurate masses and retention times of authentic standards or, when standards were not available, by using predicted retention times. p-values were adjusted for multiple testing using the Benjamini-Hochberg method. Identifications were supported by fragmentation pattern match to MzCloud database (https://www.mzcloud.org/home.aspx) and isotope distribution. The Xcalibur software package from Thermo Fisher Scientific was used for targeted peak picking and fragmentation analysis.

## Intracellular quantification of AN11736 and AN14667

Metabolism studies were performed at 1 μM AN11736 with TbWT, TbOX$^R$_A, TcoWT and TcoOX$^R$_C BSF trypanosomes. At 0 h, 1 h and 6 h timepoints $5×10^8$ parasites for *T. brucei* and $7×10^8$ parasites for *T. congolense* were collected and cell pellets resuspended in 100 μl and 50 μl 1×PBS, respectively, precipitated by addition of a 2-fold volume of acetonitrile and centrifuged at 1,700×g, 10 min at room temperature. The supernatant was diluted with water to maintain a final solvent concentration of 50% and stored at -80˚C prior to UPLC-MS/MS analysis, following a similar protocol as the one described in Wyllie and colleagues [48].

## Supporting information

**S1 Fig. Phenotype of the *T. congolense* AN11736 resistant clones.** (A) Growth curves of *T. congolense* AN11736 resistant clones B (TcoOX$^R$_B, left) and C (TcoOX$^R$_C, right) in presence (open symbols) or absence (full symbols) of AN11736 at selection concentration (12 nM for clone B and 24 nM for clone C) and of the parental wild type either cultured for a limited number of passages (TcoWT), or maintained in culture for the same time required for drug resistance selection (TcoWT_HP, high passage); doubling times for each line are indicated within brackets. (B) Stability of the resistant phenotype of clones TcoOX$^R$_B and TcoOX$^R$_C as measured by Alamar Blue assay following 90 days of culture in absence (-OX) or presence (+OX) of AN11736. (C) The resistant clone TcoOX$^R$_C retained virulence and AN11736 resistance in NIH female mice (5/5) reaching high levels of parasitaemia not cleared by treatment with 5 mg/kg of AN11736 at day 16 post infection (arrow), a dose sufficient to clear all TcoWT trypanosomes (5/5). Data in (A, B) represent means ± SD of $n$ = 3 independent biological replicates.
(TIF)

**S2 Fig. Nucleotide alignment of the nine annotated *CBP1* genes of *T. congolense*.** The alignment was made with CLC genomics workbench using the TcIL3000 reference genome available from TriTrypDB (https://tritrypdb.org/tritrypdb/).
(PDF)

**S3 Fig. Sequencing of the Cas9/sgRNA$^{CBP1}$ clone 1, a Tb427.10.1050/103 chimera.** (A) Nucleotide sequence. (B) Translated amino acid sequence. In bold, identical to only Tb427.10.1050; bold and italicised, identical to only Tb427.10.1030; normal, no difference to Tb427.10.1040; italicised and underlined are differences that are not present in 1050 or 1030. Blue and underlined, identical to Tb427.10.1050 and Tb427.10.1040. Red, underlined and bold, unique to the chimera. Underlined sequence, Cas9 targeting sequence.
(PDF)

**S4 Fig. Amino acid alignment of serine carboxypeptidases from *T. b. brucei*, *T. congolense*, *T. vivax* and *T. cruzi*.** Extract of alignment of the annotated serine carboxypeptidases from *T. congolense* (TcoCPB1A-I), *T. b. brucei* (TbCBP1A-C, Tbb427.10.1030–50), *T. vivax* (TvCBP1,

TvY486_1000990) and *T. cruzi* (TcCBP1, TcCLB.508671.20). The sequences were blasted against the entire collection of S10 carboxypeptidases stored at MEROPS Peptidase Database [42]. Family S10 has residues of the catalytic triad in the order Ser, Asp and His [29, 30] and carboxypeptidase Y (MER0002010) from *Saccharomyces cerevisiae* is the most representative gene of the family. Indicated with asterisks is the polar catalytic serine (S179) of the triad. This Ser was targeted in Tb927.10.1040 for site directed mutagenesis, substituting with a hydrophobic alanine (S179A). The alignment was made with CLC genomics workbench.
(PDF)

**S5 Fig. Isotopic distribution for fragment m/z 292.1347 (metabolite AN14667).** The fragment had an isotopic distribution that matched the simulated isotopic distribution of $C_{14}H_{19}O_5NB$ (example obtained for a TbWT replicate treated for 6 h with AN11736).
(TIFF)

**S6 Fig. Sensitivity of *T. brucei* and *T. congolense* to the AN11736 metabolite (or fragment) AN14667.** (A) No difference in susceptibility to AN14667 was found for TbWT and the resistant line TbOX$^R$_A ($EC_{50}$ 9.5 μM and 10.55 μM respectively). (B) The AN11736 resistant *T. congolense* line TcoOX$^R$_C was more than 2.5-fold more resistant to the metabolite than TcoWT ($EC_{50}$ 2.49 μM and 0.92 μM respectively); Data represent means ± SD of $n$ = 3 independent biological replicates.
(TIFF)

**S1 Table. Cross-resistance of *T. congolense* AN11736-resistant clones B (TcoOX$^R$_B) and C (TcoOX$^R$_C) to the main veterinary trypanocides and other trypanocidal compounds.**
(PDF)

**S2 Table. *In vitro* cross-resistance of *T. congolense* AN11736-resistant clones TcoOX$^R$_B and TcoOX$^R$_C to a diverse array of other benzoxaboroles.**
(PDF)

**S3 Table. Cross-resistance of *T. brucei* AN11736-resistant clones A (TbOX$^R$_A) and to a selection of trypanocides and other benzoxaboroles.**
(PDF)

**S4 Table. Primer sequences for integration into pRM481, generation of *T. congolense* tubulin locus integration plasmid and for insertion into and modification of *T. congolense* tubulin locus integration plasmid.**
(PDF)

## Acknowledgments

We thank Anne-Marie Donachie for testing activity of AN11736 on *T. congolense in vivo*, Craig Lapsley for help in DNA library production for sequencing, Suzanne Norval for advice on sample preparation for LC-MS and Stefan K. Weidt for advice on mass spectrometry fragmentation patterns.

## Author Contributions

**Conceptualization:** Federica Giordani, Isabel M. Vincent, Andrew W. Pountain, Fernando Fernández-Cortés, Ning Zhang, Liam J. Morrison, Yvonne Freund, Michael J. Witty, Rosemary Peter, Kevin D. Read, David Horn, Michael P. Barrett.

**Data curation:** Federica Giordani, Isabel M. Vincent, Andrew W. Pountain, Fernando Fernández-Cortés, Justin J. J. van der Hooft, Clément Regnault.

**Formal analysis:** Federica Giordani, Daniel Paape, Isabel M. Vincent, Andrew W. Pountain, Fernando Fernández-Cortés, Yvonne Freund, Jonathan M. Wilkes, Mark C. Field.

**Funding acquisition:** Liam J. Morrison, Yvonne Freund, Michael J. Witty, Rosemary Peter, Michael P. Barrett.

**Investigation:** Federica Giordani, Daniel Paape, Isabel M. Vincent, Andrew W. Pountain, Fernando Fernández-Cortés, Eva Rico, Ning Zhang, Darren Y. Edwards, Clément Regnault, David Horn.

**Methodology:** Federica Giordani, Isabel M. Vincent, Eva Rico, Ning Zhang, Darren Y. Edwards, Jonathan M. Wilkes, Justin J. J. van der Hooft, Clément Regnault, Kevin D. Read, Mark C. Field.

**Project administration:** Daniel Paape, Rosemary Peter, Michael P. Barrett.

**Resources:** Michael P. Barrett.

**Software:** Andrew W. Pountain, Jonathan M. Wilkes.

**Supervision:** David Horn, Mark C. Field, Michael P. Barrett.

**Validation:** Federica Giordani, Daniel Paape, Eva Rico, Liam J. Morrison, Darren Y. Edwards, Kevin D. Read.

**Visualization:** Federica Giordani, Andrew W. Pountain.

**Writing – original draft:** Federica Giordani, Daniel Paape.

**Writing – review & editing:** Federica Giordani, Isabel M. Vincent, Andrew W. Pountain, Fernando Fernández-Cortés, Liam J. Morrison, Yvonne Freund, Michael J. Witty, Rosemary Peter, Justin J. J. van der Hooft, David Horn, Mark C. Field, Michael P. Barrett.

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
