## [Decision Letter · Decision Letter 0]

13 Jul 2020

Dear Prof. Barrett,

Thank you very much for submitting your manuscript "Veterinary trypanocidal benzoxaboroles are peptidase-activated prodrugs" for consideration at PLOS Pathogens. As with all papers reviewed by the journal, your manuscript was reviewed by members of the editorial board and by several independent reviewers. The reviewers appreciated the attention to an important topic. Based on the reviews, we are likely to accept this manuscript for publication. Prior to acceptance please address the points raised by the reviewers. They are of a minor nature and no additional experimentation should be required.

Sincerely,

Dominique Soldati-Farve

Associate Editor

PLOS Pathogens

Margaret Phillips

Section Editor

PLOS Pathogens

Kasturi Haldar

Editor-in-Chief

PLOS Pathogens

orcid.org/0000-0001-5065-158X

Michael Malim

Editor-in-Chief

PLOS Pathogens

orcid.org/0000-0002-7699-2064

Reviewer Comments (if any, and for reference):

Reviewer's Responses to Questions

**Part I - Summary**

Reviewer #1: Benzoxyboroles are promising compounds to develop as drugs for treatment of trypanosome infections in animals. The rise of resistance to this type of compound is concerning. The authors present compelling evidence that resistance is due to loss of serine carboxypeptidase activity in the parasite. This implies the test compound is a prodrug.

Reviewer #2: The manuscript by Paape et al reports on the discovery that highly potent trypanocidal benzoxaborole compounds are prodrugs that are activated by serine carboxypeptidases in the trypansome cells. This is an important and significant study because it elucidates the mechanism of strong potency (and mechanism of resistance) of this particular compound class, as well as illustrating a general mechanism (prodrug processing) in trypanosomes that has relevance to other arenas of drug discovery. Through numerous experimental approaches, the investigators rigorously and convincingly demonstrate that the serine carboxypeptidases are responsible for the prodrug processing and that mutations in the genes are responsible for drug resistance. The paper is written clearly and logically.

Reviewer #3: The benzoxaboroles are an important class of trypanocides. Two molecules, SCYX-7158 (acoziborole) and AN11736, are in clinical development for human African trypanosomiasis and nagana, respectively. Paape et al. have selected T. congolense and T. brucei for resistance to AN11736. The obtained mutants were cross-resistant to other benzoxaboroles with a peptide-bond linker, and, based on whole genome sequencing, the mutants appeared to have lost paralogues of the serine carboxypeptidase (CBP) gene cluster on chromosome 10. The nature of the deletion was not further investigated, nor were expression levels of the CBP genes. Nevertheless, the authors convincingly demonstrate by reverse genetics that AN11736 is a prodrug that is cleaved by trypanosomal CBP after uptake, and that this mechanism sensitizes the trypanosomes to AN11736, presumably by creating a sink effect. I find this a conclusive and important piece of work and recommend it for publication in PLoS Pathogens.

**Part II – Major Issues: Key Experiments Required for Acceptance**

Reviewer #1: The key experiments and observations include identifying a compound fragment in wild type treated parasites that by MS conforms to a carboxypeptidase-generated fragment.

Reviewer #2: The data are thorough and adequately support their conclusions. No additional experiments are recommended.

Reviewer #3: none

**Part III – Minor Issues: Editorial and Data Presentation Modifications**

Reviewer #1: none

Reviewer #2: The Discussion did not include any mention of the natural function of the serine carboxypeptidases in these trypanosomes. It is interesting that the parasites with mutations in these genes had normal growth and ability to infect mice. Do the authors think these proteases may play a role in the procyclic life-cycle stage? Also, the final sentence of the Discussion may need further explanation. How would it help to have a different linker design with targeted activity by the trypanosome carboxypeptidases? It would probably still be subject to resistance by mutations of the peptidases in the parasites.

Reviewer #3: 1. line 116: "had increased"

2. line 128: the trypanosomes were resistant, not the compounds

3. line 428: what was the DMSO concentration?

4. figure 1, right panel: "% Survival" is not clear. Was this really a survival assay or should it read "% Growth"?

PLOS authors have the option to publish the peer review history of their article (what does this mean?). If published, this will include your full peer review and any attached files.

Reviewer #1: No

Reviewer #2: **Yes: **Frederick S Buckner

Reviewer #3: No
---

## [Editor Report · Decision Letter 1]

25 Aug 2020

Dear Prof. Barrett,

We are pleased to inform you that your manuscript 'Veterinary trypanocidal benzoxaboroles are peptidase-activated prodrugs' has been provisionally accepted for publication in PLOS Pathogens.

Best regards,

Dominique Soldati-Favre

Associate Editor

PLOS Pathogens

Margaret Phillips

Section Editor

PLOS Pathogens

Kasturi Haldar

Editor-in-Chief

PLOS Pathogens

orcid.org/0000-0001-5065-158X

Michael Malim

Editor-in-Chief

PLOS Pathogens

orcid.org/0000-0002-7699-2064
---

## [Editor Report · Acceptance letter]

23 Oct 2020

Dear Prof. Barrett,

We are delighted to inform you that your manuscript, "Veterinary trypanocidal benzoxaboroles are peptidase-activated prodrugs," has been formally accepted for publication in PLOS Pathogens.

Best regards,

Kasturi Haldar

Editor-in-Chief

PLOS Pathogens

orcid.org/0000-0001-5065-158X

Michael Malim

Editor-in-Chief

PLOS Pathogens

orcid.org/0000-0002-7699-2064